# Scarring Skin: Mechanisms and Therapies

**DOI:** 10.3390/ijms25031458

**Published:** 2024-01-25

**Authors:** Xinye Lin, Yuping Lai

**Affiliations:** 1Shanghai Key Laboratory of Regulatory Biology, School of Life Sciences, East China Normal University, Shanghai 200241, China; 51211300116@stu.ecnu.edu.cn; 2Shanghai Frontiers Science Center of Genome Editing and Cell Therapy, School of Life Sciences, East China Normal University, Shanghai 200241, China

**Keywords:** scar formation, fibroblast heterogeneity, scar treatments

## Abstract

Skin injury always results in fibrotic, non-functional scars in adults. Although multiple factors are well-known contributors to scar formation, the precise underlying mechanisms remain elusive. This review aims to elucidate the intricacies of the wound healing process, summarize the known factors driving skin cells in wounds toward a scarring fate, and particularly to discuss the impact of fibroblast heterogeneity on scar formation. To the end, we explore potential therapeutic interventions used in the treatment of scarring wounds.

## 1. Introduction

Skin injuries, including burning, wounding, surgery, and infections, always result in scars in adults. For instance, approximately 100 million individuals worldwide acquire scars on their skin due to surgery or trauma, with 15% of these scars eventually progressing into hypertrophic scars or keloids at the site of the original wounds [1]. Scars with a large surface area not only affect the aesthetics but are also accompanied by pain, itching, sclerosis, scar contracture, and other symptoms. These complications cause patients to endure social stigma and psychological trauma.

According to international clinical recommendations on scar management [2], scars can be categorized into four distinct types: mature scar, immature scar, hypertrophic scar, and keloid. Mature scars exhibit a flat appearance and share the same color as normal healthy skin, while immature scars manifest during the tissue repair stage of wound healing, displaying redness, swelling, and a slight bulge at the wound site. Over time, immature scars can either progress into normal mature scars or further develop into hypertrophic scars or keloids [3]. 

Hypertrophic scars, arising from surgical wounds or burns, are characterized by a bulging appearance at the original wound, accompanied by redness, swelling, and itching. However, when a scar extends beyond the original wound location and continues to enlarge, it transforms into a keloid. Although hypertrophic scars and keloids are clinically characterized similarly, these two types of scars exhibit histological distinctions. Keloids demonstrate irregular accumulation of type I collagen and type III collagen at the wound site, whereas hypertrophic scars display tightly arranged type III collagen with only a small amount of type I collagen [4]. 

Accumulating evidence demonstrates that scarring is a complex process; multiple cells, cytokines, and other factors are involved in this process [5]. Among these, the elaboration of fibroblast heterogeneities significantly advances our understanding of scar formation [6]. Here, we delineate the intricate process of wound healing, summarize the factors that drive skin cells in wounds toward a scarring fate, and specifically address how the heterogeneity of fibroblasts influences scar development. To this end, we discuss the potential therapeutic approaches used in scar treatment.

## 2. Wound Healing and Scars

The formation of scars usually takes place during the process of wound healing. Therefore, great efforts have been made to understand the intricate cellular and molecular processes involved in wound healing. The process of wound healing is categorized into three overlapping stages: hemostasis/inflammation, proliferation, and remodeling [7]. 

Upon injury, blood leakage occurs, and platelets aggregate to form a platelet plug followed by fibrin clot formation, which leads to hemostasis. Simultaneously, activated platelets release an array of factors, including transforming growth factor-beta (TGF-β), monocyte chemoattractant protein-1 (MCP-1), platelet-derived growth factor (PDGF), and vascular endothelial growth factor (VEGF). These factors play a crucial role in recruiting neutrophils, monocytes, and lymphocytes into the wound sites. Neutrophils, as the first cells that migrate into wound sites within the 24–48 h post-injury, engage in the phagocytosis of dead cells or invading pathogens, and subsequently release inflammatory cytokines such as IL-1 and IL-6. Following this, monocytes are recruited to wounds by components of extracellular matrix (ECM) and MCP-1, and differentiate into macrophages after 2–3 days. Macrophages also phagocytose pathogens and cellular debris and release growth factors, including PDGF and VEGF, which promote tissue granulation and repair. Throughout this period, TGF-β, PDGF, and basic fibroblast growth factor (bFGF) activate keratinocytes and fibroblasts, which facilitate the transition of wound healing from the inflammatory stage to the proliferative stage. However, it is noteworthy that the classical perspective, which posits the indispensability of neutrophils and macrophages for wound healing, has been challenged by the experimental evidence that wound healing in PU.1 null mice lacking both neutrophils and macrophages was similar to that in wild-type mice [8]. 

During the proliferative stage, the proliferation and migration of keratinocytes are required for restoring the barrier function of the epithelium. Keratinocyte growth factors (KGFs) and multiple cytokines, such as IL-17A or IL-36, induce the proliferation and migration of keratinocytes [9,10]. Fibroblasts, another major cell type activated in the proliferative stage, migrate into the wound following stimulation by PDGF, FGF, or other factors. These fibroblasts contribute to minimizing the wound surface area via the generation of contractile forces and also participate in collagen production to remodel ECM homeostasis. Approximately two-to-three weeks post-injury, the wound becomes covered by new epithelial cells and replaced with granulation tissues, which marks the transition to the remodeling stage of wound healing. During this stage, various cells that entered the wound space earlier, including fibroblasts, macrophages, and endothelial cells, undergo apoptosis or exit the wound. Type III collagen is primarily degraded to type I collagen by matrix metalloproteinases, and the orientation of collagen fibrils becomes more organized. Concurrently, the wound regains its tensile strength, gradually flattening as it undergoes structural remodeling [11]. 

Usually, wounds undergo healing within three weeks, but these quick repairs often result in scars, particularly when the wound is of significant depth. Moreover, aberrant responses exhibited by skin cells during the healing process lead to skin diseases. For example, the aberrant reactions of keratinocytes to IL-17A, IL-36, and IL-17D during wound healing contribute to inflammatory skin diseases, such as psoriasis [9,10,12]. Notably, the abnormal response of fibroblasts to TGFβ overproduction in wounds leads to the manifestation of excessive scarring or fibrosis [13]. Here we focus on elucidating the mechanisms underlying scar formation during wound healing.

## 3. Factors Involved in Driving Scar Formation

Scar formation is a complicated process. Multiple risk factors, including genetic factors, age, and certain external causes, influence fibroblast activation and play important roles in the scarring process [14,15]. In the following section, we will discuss the factors that influence scar formation.

Genetic factors. Although keloids develop in response to a dysregulated cutaneous wound-healing process, some genetic factors also predispose individuals susceptible to keloids. In 2001, Marneros’ group studied 14 pedigrees with familial keloids and found that 96 out of 341 family members exhibited keloids, which was inherited by three or five generations [16]. By using a whole genome-wide linkage search, they identified D2S1399 on chromosome 2, linked to the gene for tumor necrosis factor alpha (TNFα) inhibitory protein 6 (*TNFAIP6*) in the Japanese keloid family, and D7S499 at chromosome 7, linked to the gene for epidermal growth factor receptor (*EGFR*) in the African-American keloid family, as susceptible loci for keloid formation in two families with an autosomal dominant inheritance pattern of keloids [17]. TNFAIP6 has been reported to enhance the stability of ECM-associated proteins and promote fibroblast migration [17]. TGFβ1 is a key cytokine in the biological function of fibroblasts, and its irregularity—caused by gene polymorphisms or mutations—has been suggested to result in keloid development. For instance, several polymorphisms in the *TGFβ1* gene, such as -988C/A, -800G/A and -509C/T in the promoter region, and the mutations like Arg25Pro and Leu10Pro in exon 1, as well as Thr263Ile in exon 5, have been shown to predispose individuals to keloids [18]. In addition to genetic predisposition, DNA methylation and histone acetylation have been implicated in keloid formation. For instance, keloid fibroblasts exhibited reduced levels of histone acetylation in the secreted frizzled-related protein 1 (SFRP1), resulting in upregulation of the Wnt (Wingless; Int1) signaling pathway and subsequently promoting keloid formation [19]. All these observations suggest that genetic factors can alter the pattern of differential gene expression in keloid fibroblasts, thus predisposing individuals to keloids.

Age. Scar formation is a common consequence of skin injuries in adults, whereas fetal skin regenerates collagen fibers in neat, well-organized patterns without scar formation following injuries. Therefore, extensive efforts have been devoted to unraveling the differences between fetal and adult skin wound healing. Accumulating evidence underscores differences in cell types, ECM composition, cytokines, and growth factors between fetuses and adults [20]. Among these, fetal fibroblasts—particularly Engrailed-1 negative fibroblasts (ENFs) and CD26^+^ fibroblasts—have been identified as key effector cells in control of the transition from scar-free to scar-forming wound repair. At embryonic day 10.5 (E10.5), murine ENFs represented over 95% of total dermal fibroblasts, while Engrailed-1 positive fibroblasts (EPFs) were less than 1%. However, with age, ENFs decreased, while EPFs increased from 22.2% at E16.5, 42.9% at day 1 postnatal (P1), to 75.3% at P30. Specifically, EPFs, the primary fibroblasts in murine dorsal skin during postnatal wound healing, express and secrete collagen for connective tissue deposition [21]. Within EPFs, 94% of cells are CD26^+^ fibroblasts. CD26^+^ fibroblasts are capable of producing a dipeptidyl peptidase-IV CD26 on the cell surface in response to stimuli [22]. CD26, in turn, cleaves stromal cell-derived factor-1 (SDF-1, also known as CXCL12) to inhibit the recruitment of CXCR4-expressing mesenchymal stem cells (MSCs) [23,24,25,26], which promotes scarless wound healing by upregulating the expression of type III collagen, matrix metalloproteinases, and TGF-β3 [27]. Thus, the inhibition of CD26 by its peptidase inhibitor-diprotin A significantly reduces final scar size [21]. Moreover, fetal and adult fibroblasts synthesize and deposit distinct collagens, hyaluronic acid (HA), and other ECM components. Fetal skin and neonatal granulation tissue exhibit high expression of type III, IV, and V collagens, while adult skin predominantly expresses type I collagen [28]. The ratio of type III collagen to type I collagen in fetal skin is much higher than that in adult skin (166:0.2). The organization of type III, IV, and V collagens in fetal skin resembles a fine reticular pattern, mirroring the tissue architecture of unwounded skin and contributing to the complete restoration of skin tensile strength. Conversely, postnatal and adult skin wounds have a high deposition of thick, disorganized patterns of collagen I, resulting in increased matrix rigidity and scarring [29]. Besides collagen, HA is another important component of the ECM which is differentially expressed between fetal and adult skin [30]. HA is highly expressed in fetal skin, while it is induced in adult skin during wound healing. It contains long polysaccharide chains and its effect is dependent on the length of the polysaccharides. It has been reported that high-molecular-weight (HMW) HA suppresses the inflammatory reaction, while low-molecular-weight (LMW) HA fragments are immunostimulatory and induce the inflammatory reaction. Therefore, the positive effect of HA in fetal skin wound healing might be related to the balance between HMW and LMW HA. All these observations suggest that age is another critical factor that influences scar formation, and understanding of the phenotypic characteristics of fetal fibroblasts holds the potential to mitigate scar formation through the manipulation of these fibroblast lineages.

Gender. Females exhibit a higher propensity for developing pathological scars compared to males. A comprehensive cross-sectional analysis of a large cohort comprising 1659 patients with keloids revealed a male-to-female ratio of approximately 1:2, especially in the context of juvenile-onset keloids [31]. Although the precise mechanism by which sex affects scar formation remains unclear, clinical data suggest that hormone levels may be a major cause, as females during pregnancy have an increased likelihood of developing pathological scars [32]. This observation underscores the impact of hormone levels on the prevalence of keloids. Specifically, estrogen has been implicated as a sensitive factor in scar formation. Tamoxifen, a non-steroidal antiestrogen, has been utilized in the prevention of hypertrophic scars following surgical incision even though the efficacy of this approach has not demonstrated significant promise [33,34].

Other factors. The development of scarring is influenced by a combination of genetics, age, and sex factors, and certain anatomical factors also play important roles in scar formation. For example, scars are mostly formed in glands and cutaneous epidermal appendages such as hair follicles. Adult skin with severe damage to the basement membrane zone or deep into the dermis is more prone to scar formation after wound healing. Moreover, keloids or hypertrophic scars occur frequently in specific anatomical sites, including the scapular area, the shoulder, and the anterior chest [35]. These regions are subjected to constant or frequent skin stretching due to the body’s movements [36]. Therefore, the stretching tension of the skin, including mechanical loading and mechanical stress, emerges as another crucial factor that either promotes or aggravates the growth of keloid and hypertrophic scars following their induction. Furthermore, itching, mediated by the TGF-β-interleukin(IL)-31 axis, can intensify mechanical stress at the scar surface and promote vascular smooth muscle cell proliferation. These cascade effects subsequently lead to vascular hyperplasia and collagen deposition [37], further contributing to the development and persistence of scars. 

## 4. Fibroblast Heterogeneity and Scar Formation

Although multiple factors drive scar formation during wound healing in adult skin, fibroblasts are the major cell type responsible for this pathological process. Fibroblasts, the primary mesenchymal cell type in skin dermis, play a key role in synthesizing collagens and elastic fibers within the extracellular matrix in connective tissue and during wound healing. Their pivotal involvement spans all stages of wound healing as well as the subsequent formation of scars [38]. Although fibroblasts were previously considered to be homogenous and quiescent cells, accumulating evidence reveals their robust functional diversity across different tissues, organs, and developmental or wound healing stages [39]. The heterogeneity of fibroblasts is responsive to diverse stimuli and plays crucial roles in the intricate process of scar formation [40]. Here, we discuss how the heterogeneity of fibroblasts significantly influences the dynamics of scar formation.

### 4.1. Fibroblasts and Myofibroblasts

Before wounding, fibroblasts in adult skin are in a quiescent state, characterized by minimal cell migration and proliferation. Following injury, fibroblasts undergo a transformative process, migrating towards the wound bed and adopting a phenotypic change into proliferative and contractile cells termed myofibroblasts. These myofibroblasts, responsible for the variation in healing and scarring rates in wounded skin [41], display characteristics of both fibroblasts and smooth muscle cells and the ability to secrete collagen and contract [42]. It has been shown that myofibroblasts have three potential sources, including: (a) recruitment from bone marrow-derived fibrocytes, (b) trans-differentiation of other cell types (such as pericytes, adipocytes, epithelial cells, mesothelial cells and endothelial cells, into mesenchymal cells), and (c) proliferation and activation of resting quiescent tissue-resident fibroblasts into myofibroblasts [43]. Among these sources, the activation of tissue-resident fibroblasts has been thought of as the primary origin of myofibroblasts. The differentiation into activated myofibroblasts requires stimulation by TGF-β1 or other cytokines, or a change in the stiffness in the wound microenvironment. TGF-β1 and mechanical stress at the wound site are prominent factors in myofibroblast activation [44]. Specifically, TGF-β1 induces the expression of α-smooth muscle actin (α-SMA, also known as a biomarker of myofibroblasts), type I collagen, and a tissue inhibitor of metalloproteinase (TIMP), and then activates myofibroblasts and increases mechanical stress at the wound site [45]. This high mechanical tension further induces myofibroblast activation and then promotes myofibroblasts to release more collagen, thus increasing skin stiffness. Changes in the stiffness of the wound microenvironment also contribute to myofibroblast conversion. Myofibroblasts synthesize and deposit abundant ECM components to replace the provisional matrix as well as generate robust contractile forces that bring together the edges of an open wound, which contributes to the formation of a collagen-rich scar. Once contraction ceases, myofibroblasts undergo apoptosis and become undetectable in normal scar tissue. However, in hypertrophic scars and keloids, myofibroblasts persist for an extended period, synthesize excessive collagen I and fibronectin, and persistently express TGF-β1 and its receptors [46]. 

### 4.2. The Heterogeneity of Fibroblasts and Scar Formation

Fibroblasts are recognized for their significant plasticity and adaptability in response to environmental cues under diverse stresses and pathological conditions. Advances in single cell technologies have provided a powerful strategy to delineate phenotypic and functional subsets of fibroblasts at the single-cell level. These developments reveal the strong heterogeneity within various scar tissues and highlight that different lineages of fibroblasts play different roles during scar formation. 

SMA^+^ fibroblasts. SMA^+^ fibroblasts (also defined as myofibroblasts), characterized by their expression of α-SMA, are derived from DLK1^+^ dermal reticular fibroblasts and represent a main subtype of myofibroblasts during wound healing [47]. Recent insights derived from genetic lineage tracing and flow cytometry data have revealed that, upon induction by CD301b^+^ macrophage-derived platelet-derived growth factor C (PDGFC) and insulin-like growth factor (IGF), adipocyte precursor cells (APs) originating from En1-lineage-taced fibroblasts selectively differentiate into SMA^+^ myofibroblasts within the wound bed [48]. Over-activated SMA^+^ myofibroblasts not only increase mechanical stress at the wound site but also excessively secrete collagen. This heightened collagen production leads to ECM accumulation, ultimately culminating in the formation of pathological scars [49]. 

ADAM12^+^ fibroblasts. A disintegrin and metalloproteinase 12 (ADAM12), a membrane-anchored metalloprotease, is transiently expressed during the embryonic morphogenesis of skeletal muscles and is re-expressed in various conditions, such as liver and muscle injuries, scleroderma, and aggressive fibromatosis. ADAM12 has been shown to upregulate the expression and stability of TGF-βRII, promote the phosphorylation of the Smad2 protein, enhance the binding ability of the Smad2/3 complex to Smad4, and subsequently activate the TGF-β signaling pathway [50]. ADAM12^+^ fibroblasts, originating from a distinct proinflammatory subset of PDGFRα^+^ stromal cells, play an important role in scar formation. Following acute injury, PDGFRα^+^ stromal cells differentiate into ADAM12^+^ fibroblasts. These cells are specific progenitors of a major faction of collagen-overproducing cells responsible for scar formation. Genetic ablation of ADAM12^+^ cells or knockdown of ADAM12 has been shown to prevent mice from developing scars or fibrotic symptoms after wound healing [51], confirming the significance of ADAM12^+^ myofibroblasts as a major fibroblast subset contributing to scar formation. 

POSTN^+^ fibroblasts. Periostin (POSTN) is a member of the matricellular family of proteins that can bind to ECM and cell surface receptors. POSTN can be induced by TGF-β, IL-4 and IL-13 and play important roles in collagen fibrillogenesis by binding to type I collagen and fibronectin [52]. Moreover, POSTN is involved in the regulation of cell adhesion, proliferation, and migration through interactions with integrin receptors [53]. Notably, POSTN has been identified as a marker of mesenchymal fibroblasts with the increased expression of collagen I and III. The deletion of POSTN^+^ fibroblasts has been shown to reduce collagen production and scar formation in the hearts of mice after myocardial infarction [54]. However, it is noteworthy that high expression of POSTN has been detected in many fibrotic patients [55], and POSTN^+^ fibroblasts are markedly increased in keloids compared to normal scars [56], suggesting that POSTN^+^ fibroblasts play important roles in keloid development and may serve as a potential target for the treatment of keloid and fibrotic diseases.

Engrailed-1^+^ fibroblasts. Engrailed 1 (En1) is a transcription factor containing homeodomain, and plays important roles in the embryonic development of multiple tissues, skin wound healing, and fibrogenesis [21]. During murine embryonic development, En1 is transiently expressed in dermal fibroblasts but rapidly decreases before birth. Interestingly, En1 negative fibroblasts (ENF) can be activated to re-express En1 in response to mechanical cues within wounds [57]. In addition to mechanical signals, TGFβ has been found to reactivate En1 expression in a SMAD-dependent manner in systemic sclerosis. Overexpression of En1 in adult human dermal fibroblasts promotes fibroblast transition to myofibroblast, thus converting ENF fibroblasts into En1-positive myofibroblasts (EPF). EPFs exhibit a heightened capacity to produce ECM, which results in scar formation or tissue fibrosis.

Taken together, elucidating the heterogeneity of dermal fibroblasts advances our understanding of their contributions to skin wound healing and scar formation. This knowledge opens the door to potential developments in cell-based therapies that could accelerate wound healing with reduced scarring. 

## 5. Inflammation and Scar Formation

After wounding, a cascade of immune responses is initiated, involving multiple immune cells such as macrophages, neutrophils, dendritic cells, and mast cells. These cells are activated to secrete cytokines and chemokines [58], creating spatially and temporally limited immune responses at wound sites. These immune responses interact with fibroblasts and regulate fibroblast heterogeneity, ultimately contributing to pathological scarring (Figure 1) [59]. Notably, two key immunological drivers for scar formation are TGF-β and the type 2 cytokines, such as IL-4 and IL-13. The TGF-β family comprises three isoforms: TGF-β1, TGF-β2, and TGF-β3. Following initial injuries, TGF-β family cytokines are produced by a variety of immune cells, including macrophages and T helper cells in skin wounds, and are sustained at high levels by macrophages, fibroblasts, keratinocytes, and endothelial cells. While TGF-β1 and TGF-β2 are highly elevated in adult skin wounds, TGF-β3 is present at low levels in adult skin wounds but undergoes a significant increase in fetal skin wounds. TGF-β1 is a prototypic regulator that binds to TGFβRII, recruiting and activating TGFβRI [60]. The resulting signal transduction induces the Smad2/3 complex to enter the nuclei, and then positively regulates the expression of α-SMA, type I collagen, and tissue inhibitor of metalloproteinase (TIMP) but inhibits the expression of metalloproteases in fibroblasts (Figure 2), which promotes the activation of myofibroblasts and increases mechanical stress at the wound site for scar formation [45]. TGF-β1 also activates Smad-independent signaling pathways, such as MAPK, PI3K, and RhoA signaling pathways [61], to promote myofibroblast activation [62]. Moreover, TGF-β1-activated Smad3 recruits histone acetyltransferase CBP/P300 to regulate histone acetylation of pro-fibrotic genes and promote their expression. Sustained activation of the TGFβ1-TGFβRII-TGFβRI signaling results in long-term overactivation of fibroblasts and different subtypes of myofibroblasts. This leads to wound contraction, aberrant collagen synthesis and deposition, a higher proportion of collagen I/III, and abnormally cross-linked fiber bundles, ultimately resulting in scar formation. Therefore, TGF-β1 is considered as a key factor in regulating myofibroblast activation, scar formation, and fibrotic diseases. In contrast, TGF-β3, although signaling through the TGFβRII-TGFβRI complex, exhibits an anti-fibrotic effect and inhibits scars [63].

Similar to TGF-β1, IL-4 and IL-13 function as drivers of myofibroblast activation (Figure 1). They upregulate the expression of periostin in fibroblasts (POSTN^+^ fibroblasts) and promote the secretion of periostin, an extracellular matrix protein crucial for skin development and homeostasis. Periostin, in turn, acts on fibroblasts to induce TGF-β1 secretion via the activation of the RhoA/ROCK pathway (Figure 2). Secreted TGF-β1 further stimulates the production and secretion of periostin, thus exacerbating scar formation [64]. 

Moreover, IL-4Rα activation in macrophages by IL-4/IL-13 induces the production and secretion of resistin-like molecule beta (Relm-α). Relm-α, in turn, induces the expression of lysine hydroxylase 2 (LH2), an enzyme-directing persistent pro-fibrotic collagen cross-linking and collagen stabilization in fibroblasts. This process leads to the excessive assembly of collagen fibers and contributes to scar formation [65,66]. Besides the aforementioned cytokines, other cytokines play roles in scar formation by regulating the heterogeneity or expansion of fibroblasts [67]. For example, IL-35 has been shown to increase the expression of TGF-β1 via the activation of GP130-STAT1 or IL-12Rβ2-STAT4 signaling in fibroblasts, leading to collagen deposition and scar formation [68]. IL-6, IL-1β, IL-1α, and IL-8 are highly expressed in keloid fibroblasts, and induce the expression of TGFβ1 or render scars more sensitive to external stimuli, thus contributing to the continuous scar expansion and keloid formation [69]. Although these studies demonstrate that inflammatory cytokines play important roles in regulating fibroblast heterogeneity and functions in the formation of abnormal scars, the potential targeting of different cytokines, aside from TGF-β1, for scar treatment needs further investigation.

## 6. Therapeutic Strategies for Scar Managements

Deciphering the mechanisms of scar formation has transformed skin scar treatment into a realm where significant differences can be made in the lives of patients. A multitude of strategies, including medications, as well as genetic and cellular techniques, have undergone investigation for the prevention of scarring through topical application or injection (Table 1). Here, we delve into evidence-based literature to explore therapeutic strategies employed in the reduction and prevention of scarring conditions.

Directly targeting TGF-β1/Smad signaling. Given the well-defined role of canonical TGF-β/Smad signaling in fibroblast heterogeneity and scarring conditions, considerable efforts have been directed toward targeting this signaling pathway for scar treatment. As TGF-β3 has been found to exert adverse effects on scar formation compared to TGF-β1/2 [70], it has been used for scar treatment. In an adult rabbit skin wound experiment, the injection of TGF-β3 successfully promoted normal tissue regeneration without scarring [71]. Moreover, in a phase II clinical trial, the injection of avotermin, a human-derived TGF-β3 recombinant polypeptide, significantly improved scar appearance with fewer side effects after 12 months when administered at 200 ng/100 μL/linear centimeter wound margin once or twice [72]. In addition to TGF-β3, multiple anti-TGF-β1/2 antibodies and siRNA targeting *TGFβ/TGFβR* signaling have been explored to prevent scarring conditions. Topical application of anti-TGFβ-1 or 2 antibody demonstrated reduced collagen production in vitro [73,74] and decreased scarring in several animal models in vivo [75]. The administration of antisense phosphorothioate oligonucleotides (OGN) against *TGF-β1* and *2* in vivo significantly reduced postoperative scarring in rabbit and mouse models of glaucoma surgery [76]. Moreover, siRNA-targeting *TGFβRI* downregulated TGF-βRI expression, leading to reduced ECM deposition and a significant decrease in hypertrophic scarring in a rabbit model [77]. However, it is important to note that TGF-β3 and siRNA targeting *TGFβ/TGFβRI* have not received approval from the Food and Drug Administration for scar treatment in clinical settings. Further investigation of the practical role of the TGF family is warranted and will play a crucial role in future therapeutic protocols for scar reduction and prevention.

Indirectly targeting TGF-β signaling. In addition to directly inhibiting TGF-β/Smad signaling, certain strategies that indirectly inhibit TGF-β signaling are also used for scar treatment. For instance, intralesional injection of IFNα-2b has been shown to inhibit myofibroblast activation by downregulating TGF-β signaling. This intervention increases collagenase levels by inhibiting the secretion of metalloproteinases, an inhibitor of collagenase, thus reducing the expression of α-SMA and other ECM-related molecules [78]. Although interferon injection has achieved significant efficacy in hypertrophic scars and keloids [79], 73.7% of patients receiving interferon injection develop inflammatory complications [80], thus limiting its clinical use. Another approach involves the application of siRNA targeting TGF-β induced early gene (*TIEG*). TIEG is a protein induced by TGF-β1 and can directly inhibit Smad7 expression. Therefore, the inhibition of TIEG expression results in reduced fibroblast proliferation and migration, along with the downregulation of collagen expression [81]. Although siRNA-targeted *TIEG* has not yet been used clinically, increasing evidence suggests that RNA interference-based knockdown of key genes in the fibroblast activation pathway can promote tissue repair [82,83]. Further research is warranted to explore the clinical feasibility and safety of these indirect TGF-β signaling inhibition strategies for scar treatment.

Targeting fibroblast hyperproliferation or collagen metabolism. Hyperproliferating fibroblasts play an important role in the excessive deposition of ECM, contributing to scarring conditions. Therefore, targeting these highly proliferating fibroblasts presents another avenue for treating scarring conditions. 5-fluorouracil (5-FU), a fluorinated pyrimidine analog that inhibits nucleic acid synthesis and cell proliferation, inhibits rapidly proliferating and metabolizing fibroblasts in dermal wounds, thus reducing the excessive deposition of type I collagen and type III collagen, which occurs due to TGF-β signaling pathway activation [84]. Widely used in the treatment of hypertrophic scars and keloids, 5-FU is often administrated through intralesional injection alone or in conjunction with intralesional steroid injections or pulse dye laser treatments. This combination has shown decent efficacy in the treatment of inflamed hypertrophic scars. However, patients with keloids receiving 5-FU treatment may experience complications such as wound ulceration and hyperpigmentation [85]. Therefore, careful consideration and monitoring are necessary for assessing and managing potential adverse effects associated with 5-FU treatment in keloid patients.

Targeting collagen metabolism is another promising avenue for preventing excessive scar formation. The topical application of 1% prolyl 4-hydroxylase inhibitor post-wounding reduced the intracellular hydroxylation of proline residues necessary for collagen synthesis [86]. This intervention resulted in a decreased formation of scar tissue, as observed in a rabbit ear hypertrophic model [87]. Similarly, the inhibitor of a procollagen C-proteinase, an enzyme responsible for cleaving the C-terminal propeptide from collagen precursor to form collagen fibrils, exhibited a modest reduction in scar formation [88]. Despite these promising findings, the potential clinical use of inhibitors targeting collagen metabolism requires further investigation. It remains essential to conduct additional research to determine the safety, efficacy, and optimal application of these inhibitors in the clinical treatment of scarring conditions.

Alteration of the inflammatory response. While it is evident that an appropriate inflammatory response is essential for proper wound healing, investigations into fetal wound healing suggest that elevated inflammation levels contribute to excessive scar tissue production, which influences the ultimate outcomes of scarring. Therefore, several therapeutic approaches aim to inhibit heightened inflammation in the treatment of scarring conditions. Corticosteroid, an anti-inflammatory drug, has been widely used to treat keloid and hypertrophic scars since the 1960s. Triamcinolone (TAC), a commonly used corticosteroid, when topically applied or administered via intralesional injection, has demonstrated significant improvements in scar bulge height, flexibility, and texture, as well as a reduction in itching and pain sensation at the scar [89]. 

Nonsteroidal anti-inflammatory drugs, including cyclooxygenase-2 (COX-2) inhibitors, have also been explored for their efficacy in reducing scarring conditions. COX2 catalyzes the conversion of arachidonic acid to prostaglandin E2 (PGE2). PGE2 controls vascular permeability and reduces the infiltration of inflammatory cells. Topical application of celecoxib, a selective COX-2 inhibitor, has been shown to significantly reduce local neutrophil infiltration, PGE2 levels, and TGF-β1, thus reducing collagen deposition and scar tissue in a mouse model [90]. Despite their effectiveness, corticosteroids and COX2 inhibitors are associated with various adverse effects, including hypopigmentation around the injection site, hypervasodilation, delayed wound healing, and scar recurrence [91].

An alternative approach to curbing excessive inflammatory response in wounds involves the use of mesenchymal stem cells (MSCs). MSCs—adult pluripotent stromal cells isolated from bone marrow, adipose tissue, and umbilical cord tissue—not only inhibit inflammatory responses by suppressing TGF-β and mediating the transformation of M1 macrophages into M2 macrophages, but also restore homeostasis within scar tissue. In vivo models of hypertrophic and keloid scars with MSC transplantation have shown promising effects in improving scarring conditions. However, further investigation is needed before MSCs can be integrated in clinical practice [92].

Surgical and physical manipulation. Surgical resection stands as the predominant strategy for the treatment of keloids or hypertrophic scars. However, relying solely on surgical resection without other adjuvant treatment methods may yield a recurrence rate of scarring conditions ranging from 40% to 100% [93]. Notably, this approach can lead to the formation of larger wounds during surgery and can exacerbate the severity of keloids. To mitigate these challenges, clinical practice often incorporates steroid adjuvant therapy in conjunction with surgical resection to diminish the likelihood of substantial scarring or severe scar conditions [94].

Moreover, multiple physical manipulation techniques are employed to reduce and prevent scarring conditions. For instance, a pulsed dye laser (PDL) is used for the treatment of hypertrophic scars and keloids by specifically inhibiting vascular proliferation, a pivotal factor in scar formation [6]. Despite causing hyperpigmentation in the surrounding skin, PDL demonstrates the ability to selectively target small vessels without damaging the skin adjacent to scars. Furthermore, PDL has been observed to reduce the deposition of ECM-associated proteins by inhibiting the TGF-β signaling pathway [95,96].

Pressure therapy is another well-established physical management approach for scars, particularly in burn patients. Although its mechanism of action remains poorly understood, it is known that mechanical stress induces fibroblast differentiation into myofibroblasts, leading to increased production of fibronectin, collagens, and MMP9. Mechanical compression via pressure dressings is able to induce the expression of IL-1 and TNF, which induces cell apoptosis and scar hypertrophy regression [97]. 

Silicone-based materials represent another well-documented physical therapy for scars. As a non-invasive material, silicone gel has been widely used for hypertrophic scar treatment and scar prophylaxis. Although the mechanism of its effectiveness currently remains controversial, the prevailing theory suggests that silicone gel forms a thin film on the skin wound and promotes scar hydration [98,99]. 

In addition to the aforementioned strategies, numerous natural compounds originating from plants or animals have been used for wound resolution and scar treatment [100]. Although various strategies and natural compounds for scar treatment are available, there is no one-size-fits-all solution. Therefore, a polytherapeutic approach will be a future strategy for the reduction and prevention of scarring conditions. 

## 7. Perspectives

Despite the progress made in understanding the mechanisms underlying wound healing and scar formation, current approaches to scar management predominantly target fibroblast proliferation, collagen metabolism, or mechanical stress in wounds. These strategies have proven challenging in addressing both normal and pathological scars, thus making effective treatment and prevention. Most recent discoveries of the heterogeneity of dermal fibroblasts highlight the role of distinct dermal fibroblast lineage in skin scar formation after injury. Therefore, direct interventions towards this specific dermal fibroblast lineage hold promise for scar treatment. Moreover, insights into the mechanisms of skin regeneration during fetal development could pave the way for scarless treatment options in adults. However, the precise mechanism response for scarless repair in fetal skin remains elusive. More efforts are required for understanding mechanistic differences in wound healing between scarless fetal wounds and scarring adult wounds. Moreover, the development of a mouse or rodent model or a 3D skin culture that can accurately resemble human skin wound healing is imperative, given the heterogenic disparities between human and mouse biology. To this end, we anticipate that the aforementioned investigations will ultimately lead to the development of novel therapies capable of facilitating scarless wound healing. 

## Figures and Tables

**Figure 1 ijms-25-01458-f001:**
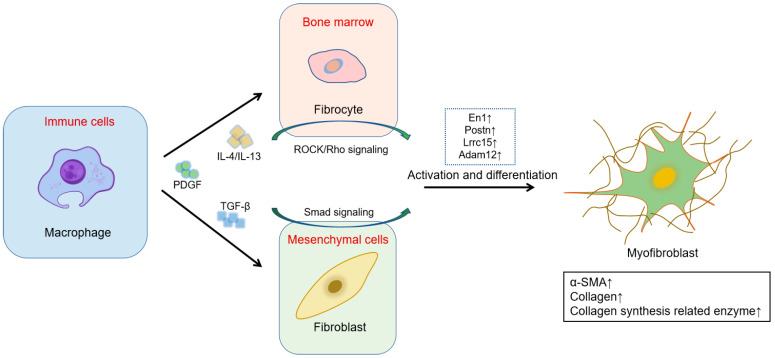
Cross-talk between fibroblasts and immune cells. Upon injury, immune cells, such as macrophages, are activated to release cytokines, including PDGF, TGF-β, and IL-4/IL-13. IL-4/IL-13 acts on fibrocytes and induces fibrocytes differentiation into fibroblasts, such as Postn^+^ fibroblasts, via the activation of ROCK/Rho signaling. TGF-β acts on fibroblasts and promotes fibroblast differentiation into different lineages of myofibroblasts, such as En1^+^, Lrrc15^+^, or Adam12^+^ fibroblasts. The activated fibroblasts and myofibroblasts subsequently generate elevated levels of collagens and other ECM components that play a pivotal role in the process of scar formation. Arrows demonstrate the levels of En1^+^, Postn, Lrrc15^+^, or Adam12^+^ are increased.

**Figure 2 ijms-25-01458-f002:**
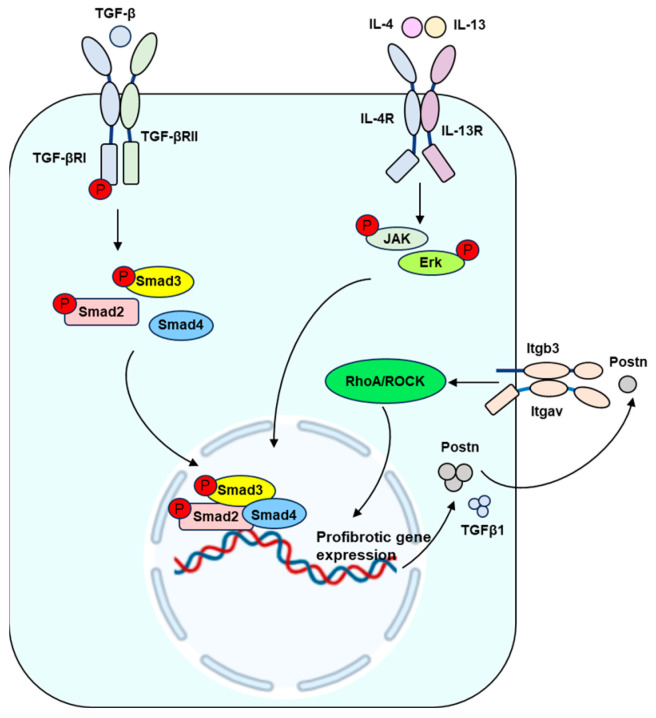
Signaling pathways involved in the induction of profibrotic gene expression. TGF-β1 binds to TGFβRII and then recruits and activates TGFβRI. The resulting signal transduction induces the phosphorylation of Smad2 and Smad3. Phosphorylated Smam2 and Smad3 complex with Smad4 to enter the nuclei, and then positively regulate the expression of Postn, α-SMA, type I collagen, and a tissue inhibitor of metalloproteinase (TIMP) but inhibits the expression of metalloproteases in fibroblasts. IL-4 and IL-13 bind to their receptor IL-4R and then activate JAK and Erk, which results in the expression of periostin (Postn) in fibroblasts. Postn, in turn, acts on fibroblasts to induce TGF-β1 secretion via the activation of the Itgb3/Itgav-RhoA/ROCK pathway. Secreted TGF-β1 further stimulates the production and secretion of periostin and other profibrotic genes, thus exacerbating scar formation.

**Table 1 ijms-25-01458-t001:** Different treatments of scars in clinical trials.

Treatment	Target	Advantage	Disadvantage
IFN injection	Inhibit the activity of matrix metalloproteinases; Reduce the deposition of collagen	Effective	Expensive; Have inflammatory complications
TGF-β3 injection	Inhibit fibroblast differentiation	Safety; Less side effects	Less effective
Stem cell therapy	Inhibit TGF-β; Inhibit inflammatory responses	Convenient	Need more clinical data
RNA-based therapy	Inhibit fibroblast proliferation and activation	Precise	Need more clinical data
Surgical resection	Scar removal	Widely used	High recurrence rate
Pressure garment therapy	Reduce collagen synthesis; Induce cell apoptosis	Relieve scar-associated itch and pain	Inconvenient
Silicone-based therapy	Inhibit excess capillary regeneration; Reduce collagen deposition	Flexible; Non-invasive	Less effective
Intralesional corticosteroid injection	Inhibit collagen synthesis and fibroblast proliferation	Improve scar texture; Reduce itch and pain	High recurrence rate
Fluorouracil injection	Inhibit the activity of high proliferating fibroblast	Reduce recurrence rate	Need more clinical data
Photodynamic therapy	Inhibit collagen synthesis and fibroblast proliferation	Instant effect; Effective	Need more clinical data
Botulinum toxin injection	Promote normal tissue remodeling	Inhibit scar expansion	Need more clinical data
Cryotherapy	Microvascular damage and ischemic necrosis	Effective	Long treatment cycle
Fat transplantation	Inhibit fibroblast proliferation	Easy to obtain; Cheap	Need to improve graft cell survival rate
Platelet-rich plasma	Promote wound healing and collagen remodeling	Convenient	Need more clinical data
Laser therapy	Reassemble collagen fibers into normal collagen fiber structure	Selectively influence collagen fibers without damaging the surrounding skin	Expensive; Have side effects

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
