# Peer review of "Scarring Skin: Mechanisms and Therapies"

_ijms, 2024, doi:10.3390/ijms25031458_

Round 1
Reviewer 1 Report
Comments and Suggestions for Authors
This is a very nice review manuscript describing newest developments in understanding of pathophysiology of scarring, and real and potential therapeutic options. I have a very few minor comments/questions:
Paragraph on Genetic factors, starts om line 98: overexpression of TGFβ and FGF in scars is well documented (described later in this study), is there a genetic component to it, at least in some cases?
Next paragraph: Age, starts n line 115. Does hyaluronic acid (or hyaluronan) contribute to scarless healing in prenatal life and early life? Is it produce in higher amounts than in adults?
4.2. The heterogeneity of fibroblasts…: starting on line 211. Are there any biomarkers for myofibroblasts?
6. Therapeutic strategies…starts on line 328: it is advisable to mention in the first paragraph of this section that the treatment is topical, either by injections or as a cream/lotion.
Comments on the Quality of English LanguageThough the review is written in very good English, the authors need to go over the paper to correct minor mistakes.
Author Response
Reviewer 1:
Comments and Suggestions for Authors
This is a very nice review manuscript describing newest developments in understanding of pathophysiology of scarring, and real and potential therapeutic options. I have a very few minor comments/questions:
Comment: Paragraph on Genetic factors, starts om line 98: overexpression of TGFβ and FGF in scars is well documented (described later in this study), is there a genetic component to it, at least in some cases?
Response: Thank you for the reviewer’s thoughtful suggestion. We have integrated additional evidence regarding the association between TGFβ1 gene polymorphisms or mutations and keloid development in the section on “Genetic factors”, as indicated in lines 115-120.
Comment: Next paragraph: Age, starts in line 115. Does hyaluronic acid (or hyaluronan) contribute to scarless healing in prenatal life and early life? Is it produce in higher amounts than in adults?
Response: Yes, hyaluronic acid has been suggested to contribute to scarless healing in prenatal life and early life. We have included the information on its roles in scarless healing in the “Age” section, specifically detailed in lines 159-167.
Comment: 4.2. The heterogeneity of fibroblasts…: starting on line 211. Are there any biomarkers for myofibroblasts?
Response: Yes, α-SMA is widely recognized as a biomarker for myofibroblasts. The information has been added in the “fibroblast and myofibroblasts” section, specifically at line 230.
Comment: 6. Therapeutic strategies…starts on line 328: it is advisable to mention in the first paragraph of this section that the treatment is topical, either by injections or as a cream/lotion.
Response: As the reviewer suggested, we have included the information regarding the treatment by topical application or injection. This addition can be found in the text at line 374.
Comments on the Quality of English Language
Comment: Though the review is written in very good English, the authors need to go over the paper to correct minor mistakes.
Response: We appreciate the reviewer’s feedback regarding the language. We have thoroughly reviewed the paper and corrected all the typos and misleading sentences.
Reviewer 2 Report
Comments and Suggestions for Authors
The authors described "Scarring skin: Mechanisms and therapies" in review style. This topic should be informative for potential readers. However, there is less what is new. A lot of papers related to scarring skin and therapies have been already published (e.g. Shirakami E, et al. Burns Trauma, 2020, Lee HJ, et al. IJMS, 2019,,,). So, please add more new things to this manuscript (e.g. Figure describing interaction of molecular).
Comments on the Quality of English LanguageModerate editing of English language is required.
Author Response
Reviewer 2:
Comment: The authors described "Scarring skin: Mechanisms and therapies" in review style. This topic should be informative for potential readers. However, there is less what is new. A lot of papers related to scarring skin and therapies have been already published (e.g. Shirakami E, et al. Burns Trauma, 2020, Lee HJ, et al. IJMS, 2019,). So, please add more new things to this manuscript (e.g. Figure describing interaction of molecular).
Response: We appreciate the reviewer’s valuable input regarding the need for more recent and innovative content in the manuscript. In this revised version, we have included citations of some latest works and introduced Figure 2 to illustrate the signaling pathways involved in the induction of profibrotic gene expression. We apologize for any oversight and appreciate the opportunity to enhance the manuscript’s quality.
Comments on the Quality of English Language
Comment: Moderate editing of English language is required.
Response: We thank the reviewer for the feedback on the need for moderate editing of the English language. We have thoroughly reviewed the manuscript, corrected all the typos, and clarified any misleading sentences.
Round 2
Reviewer 2 Report
Comments and Suggestions for Authors
The authors revised the manuscript precisely. Thank you for this opportunity.